# Microstructure and Mechanical Properties of High Vacuum Die-Cast AlSiMgMn Alloys at as-Cast and T6-Treated Conditions

**DOI:** 10.3390/ma12132065

**Published:** 2019-06-27

**Authors:** Fei Liu, Haidong Zhao, Runsheng Yang, Fengzhen Sun

**Affiliations:** 1National Engineering Research Center of Near-net Shape Forming for Metallic Materials, South China University of Technology, Guangzhou 510641, China; 2Department of Mechanical Engineering, Imperial College London, Exhibition Road, London SW7 2AZ, UK

**Keywords:** T6 heat treatment, AlSiMgMn alloys, Fracture, In-situ SEM observation, Microstructure

## Abstract

Al–Si–Mg based alloys can provide high strength and ductility to satisfy the increasing demands of thin wall castings for automotive applications. This study has investigated the effects of T6 heat-treatment on the microstructures, the local mechanical properties of alloy phases and the fracture behavior of high vacuum die-cast AlSiMgMn alloys using in-situ scanning electron microscopy (SEM) in combination with nano-indentation testing. The microstructures of the alloys at as-cast and T6 treated conditions were compared and analyzed. It is found that the T6 heat treatment plays different roles in affecting the hardness and the Young’s modulus of alloy phases. This study also found that the T6 heat treatment would influence the failure modes of the alloys. The mechanisms of crack propagation in the as-cast and T6 treated alloys were also analyzed and discussed.

## 1. Introduction

High pressure die-casting (HPDC) has been considered as a simple but effective method for the fabrication of aluminum alloy parts [1,2]. Due to its advantages of high efficiency and short production cycle, HPDC has been widely employed by automotive industries. Among aluminum-silicon and aluminum-silicon-magnesium based alloys, AlSiMgMn alloys, designated as AA365 (Silafont 36^®^), usually have good castabilities and excellent mechanical properties [3], and therefore, they are widely used to make automotive structural components by HPDC process [4,5,6,7]. In the conventional die-casting process, the high velocity injection of molten metal unavoidably entraps a certain amount of cavity gas in the material, which results in the formation of porosities in the castings [8,9,10,11]. The porosities are the potential initiation sites of fracture and fatigue failure, which are detrimental to the casting properties and performance [12,13].

In HPDC process, high vacuum technology can significantly reduce porosities by avoiding gas entrapment [14,15] and increase the casting integrity. Moreover, the mechanical properties of the castings can be further improved by heat-treatment. However, the mechanical properties of high vacuum die-cast AlSiMgMn alloys in actual production show considerable variations. Therefore, several researchers have studied the factors affecting the properties of AlSiMgMn alloys. Fernández [16] found that increasing the solution temperature and time could enhance the yield and ultimate tensile strength of a new secondary alloy-AlSi10MnMg (Fe) produced using vacuum assisted HPDC. When the solutionizing temperature increases from 475 °C to 525 °C, the strength and the hardness of an artificially aged HPDC AlSi7MgMn increase markedly [17]. Srivastava [18] reported that the heat-treated AlSi9MgMn alloys stored for one year (naturally aged) were associated with higher yield strength due to the precipitation of Mg_2_Si particles, compared to the as-cast material. He also found that heat treatment was expected to lead to better properties through the spheroidization of Si particles and subsequent age hardening due to the presence of Mg in the alloy [19]. However, studies of the influence of heat treatment on the fracture behavior of the high vacuum HPDC AlSiMgMn alloys are very limited [20,21]. The in-situ observation has been successfully applied to investigate the crack initiation and propagation in castings [22,23], as it enables researchers to experimentally correlate the microstructures of material with the mechanical properties during the entire loading process. However, the application of in-situ observation of the tensile failure process of the high vacuum HPDC AlSiMgMn alloys has not been found to date.

The aim of this paper is to reveal the effects of heat treatments on the different alloy phases and mechanical behavior of AlSiMgMn material produced by high vacuum HPDC. The microstructures of the high vacuum HPDC AlSiMgMn alloys at the as-cast and T6 heat treatment conditions were investigated by scanning electron microscopy (SEM) and energy dispersive spectroscopy (EDS). The hardness and Young’s moduli of different phases were measured with nano-indentation technique. Tensile testing of the alloys was conducted with in-situ SEM observation. By investigating the interplay between the microstructures and the crack paths in the alloys, the effects of T6 heat treatment on the fracture behavior of the alloys were analyzed and discussed.

## 2. Materials and Methods

### 2.1. Materials

In this work, the investigated AlSiMgMn alloys were from practical automotive shock tower components fabricated with Buhler 3500 tons cold chamber die-casting machine integrated with high vacuum units. The chemical compositions of the alloy were analyzed using ARL4460 and were presented in Table 1. In the die-casting process, the pouring and die temperatures were 650 °C and 200 °C, respectively. The pressure of die-cavity in producing was 35~45 mbar, which satisfies the requirement of critical pressure of high vacuum, 50 mbar. 

### 2.2. T6 Heat Treatment

Heat treatment experiments were performed in an electrical resistance furnace (Guoju Furance Company, Luoyang, China) with a control accuracy of ±1 °C. Two steps of solution heat treatments were carried out. Firstly, the alloys were solution heat treated at 470 °C for one hour, and then the alloys were treated at 485 °C for another 2 h. After the solution heat treatment, the samples were immediately quenched in cold water (20 °C) and then were artificially aged at 170 °C for 3 h in a fan-circulating furnace (Guoju Finance Company, Luoyang, China), followed by ambient air cooling.

### 2.3. Microstructure Analysis

Optical microscope (DMI 5000, Leica Company, Solms, Germany) was adopted to investigate the microstructure of untreated and treated samples. The in-situ observation was performed with a Shimadzu SEM-Servopluse test machine (Shimadzu Corporation, Kyoto, Japan) under ambient temperature for observing the crack paths and analyzing the microstructure.

### 2.4. Nanoindentation Test 

Prior to the indentation tests, the sample was polished by 400, 800, 1500, 2000 grit SiC abrasive papers and rinsed with distilled water and ethanol (Fuyu Fine Chemical Company, Tianjin, China). All the specimens were indented to a depth of 1000 nm at a constant rate of 0.1 nm/s, with the maximum load remaining for 2 s and unloading with the same rate. Nanoindentation was performed in 8–10 randomly selected points on different phase surfaces.

The criteria for hardness *H* and modulus *E_r_* are calculated by following Equations (1) and (2), which were proposed by Oliver and Pharr [24]:(1)$$H=\frac{P_{max}}{A_{c}}$$
(2)$$E_{r}=\frac{S\sqrt{\pi }}{2\beta \sqrt{A_{c}}}$$
where *P_max_* is the applied maximum load, *A_c_* is the projected contact area of the indenter, *S* is the contact stiffness given by Equation (3) and *β* is the constant (about 1.034).
(3)$$S={\left(\frac{dP}{dh}\right)}_{h_{max}}$$
where *h* is the indentation depth. The project contact area *A_c_* is obtained from following relations:(4)$$A_{c}=24.5h_{c}^{2}$$
(5)$$h_{c}=h_{max}-\varepsilon \frac{P_{max}}{S}$$
where *h_c_* is the contact depth, and *h_max_* is the indentation depth when the applied load is maximum.

### 2.5. Tensile Test 

The samples were prepared by wire cut electrical discharge machining (WEDM) (Ronyan Machinery Company, Dongguan, China) according to the dimension in Figure 1. In addition, in the middle of the sample, a v-notch was processed on one side with angle of 60° and depth of 0.3 mm. The surfaces of tensile sample were polished by 2000 grit SiC abrasive papers, followed by a rinse with distilled water and ethanol, to remove possible sites of stress concentration and ensure a smooth surface in the in-situ tensile experiment. Then, these samples were etched by 0.5% hydrofluoric acid (Huantai Huiyong Chemical Company, Zibo, China) for better observation on the in-situ tensile experiment. The tensile speed was set at 1 μm/s for observing the fracture process. During the tensile test, the images of sampling surfaces were recorded at each propagation of 0.3 mm. Five as-cast samples and T6 heat treatment samples were conducted the in-situ tensile tests. In this paper, two representative samples for the as-cast and T6 heat treatment states are selected for analysis. The fracture was analyzed through SEM (Nova Nanosem 430, FEI Company, Hillsboro, USA) equipped with an energy dispersive spectroscope (EDS) (FEI Company, Hillsboro, USA) device for identifying the structural components.

## 3. Results and Discussion

### 3.1. Microstructure

Figure 2a–d show the microstructures of AlSiMgMn alloys in the as-cast and T6 conditions, respectively. There are nearly no porosities for both conditions, which confirms the validity of high vacuum used in the die-casting process. In Figure 2a,b, the α-Al grains, intermetallics, and eutectics are observed in the as-cast alloys. The morphology of eutectic Si particles is fibrous or short rod-like, and fewer Mg_2_Si distributes on the α-Al matrix in Figure 2b because of the low-magnesium (<0.1 wt. %) of AlSiMgMn alloys. In Figure 2c,d, the α-Al grains, intermetallics and eutectics are also identified in the T6 alloys. The eutectic Si particles are clearly fragmented, and spherical Si particles are also observed, see Figure 2c. The Mg_2_Si is rarely observed in Figure 2c,d. This is because the compounds dissolved into α-Al matrix during the solution process. The EDS analysis in Figure 2e proves that the intermetallics are α-Fe phase (Al_15_(FeMn)_3_Si), which has been also observed in other studies [25,26]. However, the Fe-rich intermetallics were not significantly affected by the T6 heat treatment as shown in Figure 2d, compared to Figure 2b. 

### 3.2. Nano-Indentation Hardness Test

Figure 3 shows the load-displacement curves conducted on different phases of the alloy. There are obvious differences in the indentation response among the three phases. The quantitative results of the Young’s modulus and hardness of the phases in the as-cast and the T6 alloys are shown in Table 2. The discontinuity pop-in events of α-Fe intermetallic’ curves were observed during the loading as shown in Figure 3. According to the literature [27], microcracks might occur in the brittle α-Fe intermetallics, which causes small pop-ins and discontinuities in the load-displacement curve. Compared to the as-cast alloys, the hardness values of the T6 alloys decrease remarkably. The hardness of eutectics decreases from 145.2 to 98.1 HV. This is because the Mg, Si of Mg_2_Si particles and eutectic Si particles dissolved into the α-Al matrix [28] after T6 heat treatment. The hardness of α-Fe intermetallics decreases from 430 to 167 HV after the treatment. The work in [29] pointed out that the stress concentration on α-Fe intermetallics could be reduced after T6 heat treatment, which may lead to hardness reduction. Table 2 also shows that the values of Young’s modulus of α-Fe intermetallics and eutectics in the as-cast alloys are similar, but that of α-Al grains is lower. After T6 heat treatment, the Young’s modulus of α-Al and α-Fe phases increases to 85.1 and 102.1 GPa, respectively, while the modulus of eutectics decreases to 79.9 GPa. This is attributed to the dissolving of Mg_2_Si and Si particles from eutectics into α-Al matrix during the solution treatment. 

### 3.3. Tensile Behavior 

#### 3.3.1. As-Cast Alloys

Figure 4 shows the nominal tensile stress-strain curve of the as-cast alloy. The sudden drops in the stress-strain curve were caused by the suspending of loading in in-situ tensile test to record the sample damage. The ultimate tensile strength (UTS) and elongation of as-cast alloys are 265 MPa and 3.8%, respectively. The stress-strain curve can be divided into three stages: the elastic stage, the crack initiation stage and the ultimate fracture stage. Plastic deformation occurred in the second stage. As the stress approaches the UTS, the alloy instantly broke in the third stage. To understand the effect of microstructure on the tensile failure process, five representative stops were selected to discuss the crack changes in the later part. Point a represents a moment in the elastic stage, point b represents a moment in the crack initiation stage and points c-e represent moments in the ultimate fracture stage as shown in Figure 4.

Figure 5 displays crack initiation and propagation process of the as-cast alloy. Slip-bands formed in large α-Al grains and microcracks initiated in α-Fe phases. The α-Al grains underwent apparent plastic deformation in Figure 5a. Since the α-Al grains have lower Young’s modulus and hardness (see Table 2), it deformed earlier than the eutectic phases. Under tension, the α-Al grain elongated along the loading direction and contracted in the transverse direction. Therefore, many sites of the lateral side where α-Al grains present were concaved due to the contraction and the surface became uneven, as seen in in Figure 5b. 

Figure 6 shows the crack initiation and propagation of the as-cast alloy at different loads. Some round hydrogen pores on the sample surface can be observed in Figure 6a, which is consistent with the results of literature [30]. At the second stage as shown in Figure 6b, microcracks appeared on the sample. As shown in Figure 6b,d,e, the microcracks with length of 3~6 microns indicated by red rectangles were found in the α-Fe intermetallics. Microcracks in the eutectic region were also found as marked by the red circles. With increasing load, the concave of α-Al grain surface became more obvious due to their plastic deformation, as shown in Figure 6b. 

In the experiment, after the crack initiated, the main crack propagated rapidly in the last stage, as shown in Figure 6c–e. As a result, only three images were captured for this stage. Comparing Figure 6a,c, it can be deduced that the main crack initiated from the hydrogen pore near the notch. The number of microcracks increased with the load increasing as shown in Figure 6d. Figure 6e shows that some microcracks initiated adjacent to the main crack. However, the orientations of the microcracks were not perpendicular to the loading direction, indicating the difference between the internal stress and the load direction. To further check the effect of phases on crack propagation, the elements distribution near main crack were analyzed with EDS. Figure 6f gives the Si element distribution in the region marked with blue rectangle in Figure 6e. In this selected region, many Si particles with fibrous or rod-like shapes are found at the as-cast condition. Figure 6e,f reveals that the main crack quickly passed through the eutectic Si regions during crack propagation. This fracture behavior is consistent with the result in [12], indicating that the fibrous or rod-like eutectic Si particles are prone to break under tensile loading. 

Figure 7 presents fracture morphology of the as-cast alloy. Some microcracks exist on the fracture surface as displayed in Figure 7a. Combined with the EDS analysis results in Figure 7b,d, the existence of α-Fe intermetallics and eutectics are confirmed. The cleavage facets and brittle fracture features are also shown in Figure 7a,c. 

In summary, for the as-cast high vacuum HPDC AlSiMgMn alloys, the α-Al grains have the lowest modulus and deformed earlier than other phases. As a result, the α-Al grain surfaces concaved. With the increasing stress, microcracks formed on α-Fe intermetallics and eutectic regions due to the presence of fibrous eutectic Si particles. The main crack propagated through the eutectic regions, leading to a brittle fracture event.

#### 3.3.2. T6-Treated Alloy 

Figure 8 presents the nominal tensile stress-strain curve of the T6-treated specimens. The first two stages of the treated specimens are similar to those of the as-cast alloys. When the stress reaches the UTS (point c in Figure 8), the strain continues to increase with loading. It indicates that the crack propagated more slowly than the as-cast alloys in the last stage. The elongation to failure has been significantly improved, up to 9.1%. Compared to the result in Figure 4, it is noticed that the alloys gain more ductility after T6 treatment. In the T6 heat treatment of cast AlSi alloys, spherical eutectic Si particles forms, and the average spacing between eutectic Si particles increases [18]. Therefore, the spherical eutectic Si particles after the T6 heat treatment in the alloys (Figure 2) enhance the ductility. However, the UTS decreases to 224 MPa compared to the value of 270 MPa in the as-cast alloys. This strength reduction is because the total fraction of eutectic decreased after T6 treatment, as shown in Figure 2d, and the preparticipation of Mg_2_Si during the aging was insufficient due to the low Mg content in the raw material. 

As shown in Figure 9a, microcracks initiated at small subsurface pores near the notch root with applied loading. This is similar to the as-cast alloys. In the T6 alloys, more slip bands formed near the main crack than the as-cast alloys. Many voids initiated between eutectic regions and α-Al grains, as indicated in Figure 9b. Figure 9c shows that the final crack path was oriented with a relatively larger angle with respect to the tensile direction compared to that in the as-cast alloy (Figure 5b). It also suggests that the crack propagated mainly through the α-Al matrix, as indicated by the arrows.

As shown in Figure 8, point a represents the tensile response in the first stage, point b in the second stage, points c-d in the third stage and points e-h in the last stage. Figure 10 presents the detailed information of the crack initiation and propagation in the T6 alloys. The hydrogen pores and machining unevenness existed near the notch root as shown in Figure 10a. The tensile test was performed by gradually increasing the applied load. Two micro cracks were clearly observed on the notch root, as shown in Figure 10b. From Figure 10a, it is found that the upper and lower cracks initiated from machining roughness and subsurface hydrogen pore, respectively. Then, these cracks slowly propagated before the applied loading reaching UTS. When the applied load exceeded 655 N (see Figure 10c), the lower crack continued to propagate, but the upper one stopped. Although many α-Fe intermetallics were identified in the front of the main crack tip, only relatively larger α-Fe intermetallics with a size over 15 microns broke as shown in Figure 10c. In contrast, microcracks in the as-cast alloys formed at the α-Fe intermetallics, which have quite smaller sizes in Figure 6b,d. This is due to releasing stress concentration of the intermetallics after T6 heat treatment [29]. As shown in Figure 10d–f, the concave of α-Al grains was not obvious compared to the as-cast sample. This is considered to be caused by two reasons. The first one is that the modulus of eutectic regions decreased after the T6 treatment and indeed is the lowest (Table 2), so that the eutectics firstly deformed. The second reason is that the α-Al grains are strengthened due to participation during the ageing. Figure 10d shows that many voids formed in the eutectic regions because of the difference in plasticity between the Si particles and α-Al grains. Figure 10d also indicates that the breakage of Si particles took place.

As the tensile deformation continuously increased, the main crack propagated quickly in the final stage. Figure 10e shows that the crack propagated at approximately 45 degrees relative to the loading axis (580 N). It implies that the sample fractured in a shear dominant mode, leading to more slip bands (see Figure 10f). Figure 10g,h shows that during the crack propagation, many voids formed in eutectics and plastic deformation occurred in the α-Al grains. The main cracks propagated forward by connecting the voids and passing through the grains.

The fracture morphology and EDS analysis of the T6 alloys were given in Figure 11. Eutectic Si regions were not obviously observed on the fracture surface (Figure 11a) compared to the as-cast alloys in Figure 7c. This is also confirmed by EDS analysis in Figure 11b. The α-Fe intermetallics were observed on the surface, as indicated in Figure 11c,d, and a lot of dimples were found on the fracture surface. These results confirm that the main crack propagated forward by shearing α-Al grains. After the T6 heat treatment, the alloy fracture changed from a brittle mode to a more ductile mode.

According to the results in this study, in application of the alloys in foundry producing, the contents of Fe and Mn should be strictly controlled to avoid large α-Fe intermetallics, which promote formation of microcracks and fracture, and the solution treatment is suggested if high ductility is required for castings with the alloys.

## 4. Conclusions 

It this study, the effects of T6 heat treatment on the microstructures, the mechanical properties of the alloy phases and the fracture behaviors of AlSiMgMn alloys produced with high vacuum HPDC method were experimentally investigated using nanoindentation and in-situ SEM observation. The main conclusions are as follows:

(1) The alloy microstructure consists of α-Al grains, intermetallic compounds Al_15_(Fe,Mn)_3_Si_2_ and Al-Si eutectics. After the T6 heat treatment, the hardness of all alloy phases significantly decreased. The Young’s modulus of the α-Al grains and α-Fe intermetallics increased whereas the modulus of the eutectic regions decreased. The T6 treatment decreased the tensile strength of the alloy improved elongation.

(2) In the as-cast AlSiMgMn alloys, microcracks formed in the α-Fe intermetallics and eutectic regions due to the presence of fibrous eutectic Si particles. The main crack propagated quickly through the eutectic regions, leading to brittle failure.

(3) In the T6-treated AlSiMgMn alloys, microcracks were only formed near relatively larger α-Fe intermetallics. The crack propagated by connecting the voids formed between the α-Al matrix and Si particles due to their differences in plasticity. The crack propagated through α-Al grains with shearing, exhibiting more ductile fracture behavior compared to the as-cast alloys. 

## Figures and Tables

**Figure 1 materials-12-02065-f001:**
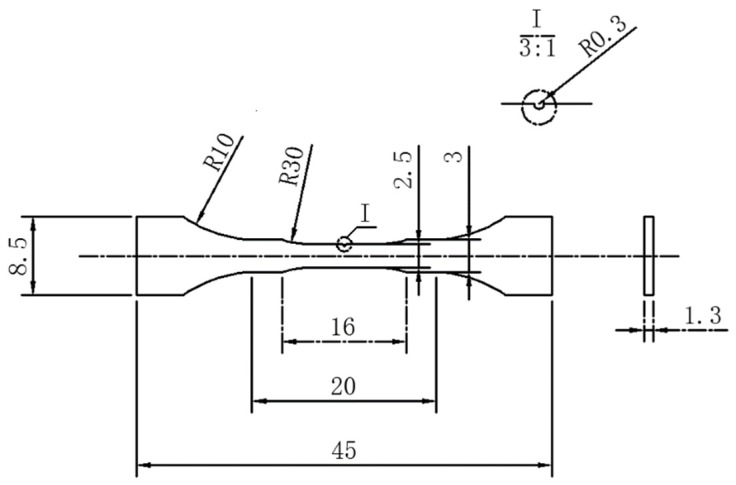
Dimensions of in-situ tensile samples (mm).

**Figure 2 materials-12-02065-f002:**
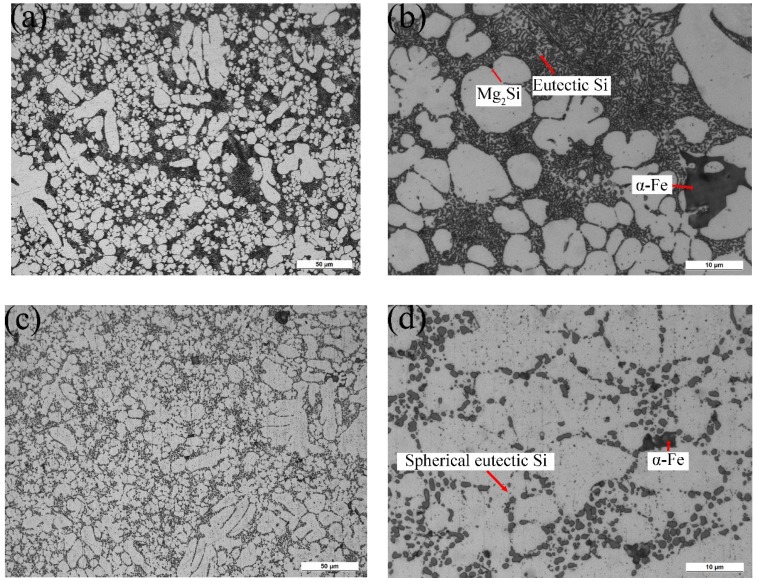
The microstructure of AlSiMgMn alloy. (**a**,**b**) As-cast; (**c**,**d**) T6 heat treatment; (**e**) energy dispersive spectroscopy (EDS) analysis results of α-Fe intermetallics.

**Figure 3 materials-12-02065-f003:**
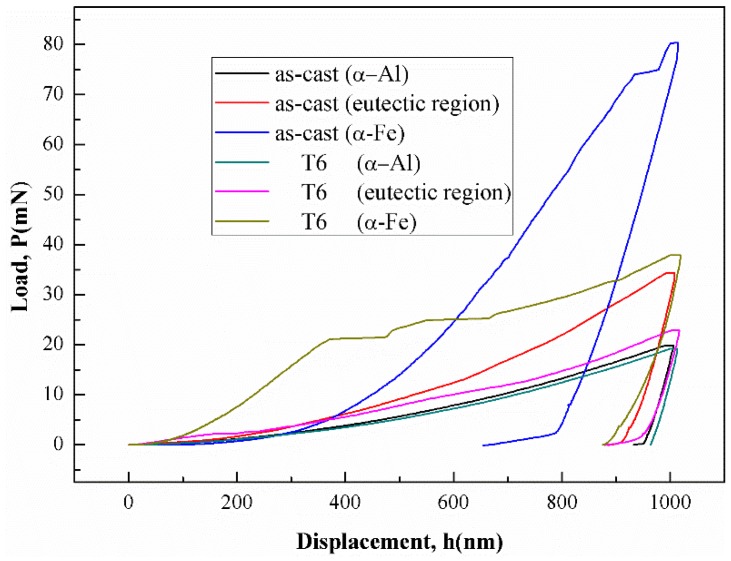
Load-displacement curves of nano-indentation testing of as-cast and T6 AlSiMgMn alloys.

**Figure 4 materials-12-02065-f004:**
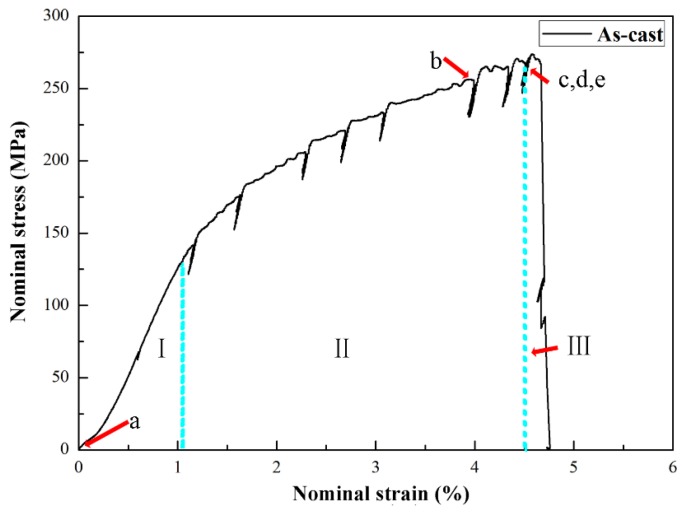
In-situ nominal tensile stress-strain curve of the as-cast alloy.

**Figure 5 materials-12-02065-f005:**
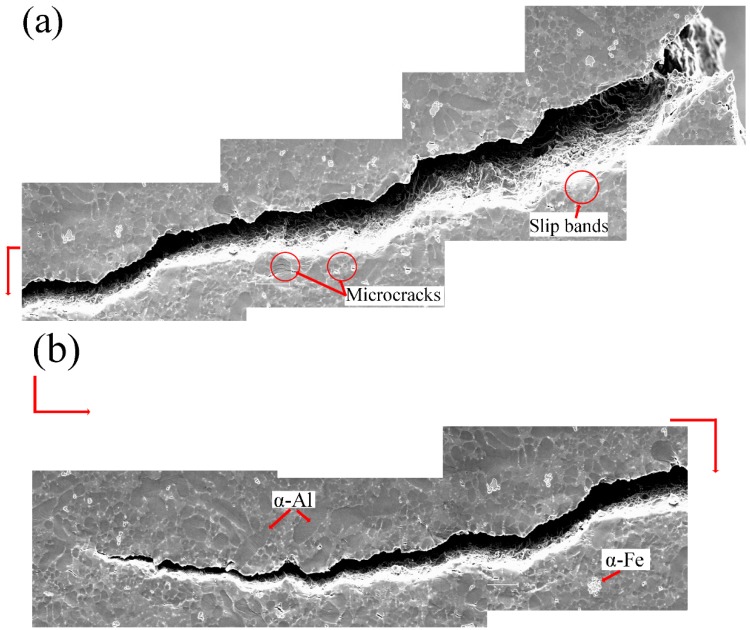
Crack propagation during in-situ tensile testing the as-cast alloy, showing (**a**) formation of slip-bands and microcracks initiation in α-Fe phases, (**b**) concaving of α-Al grains.

**Figure 6 materials-12-02065-f006:**
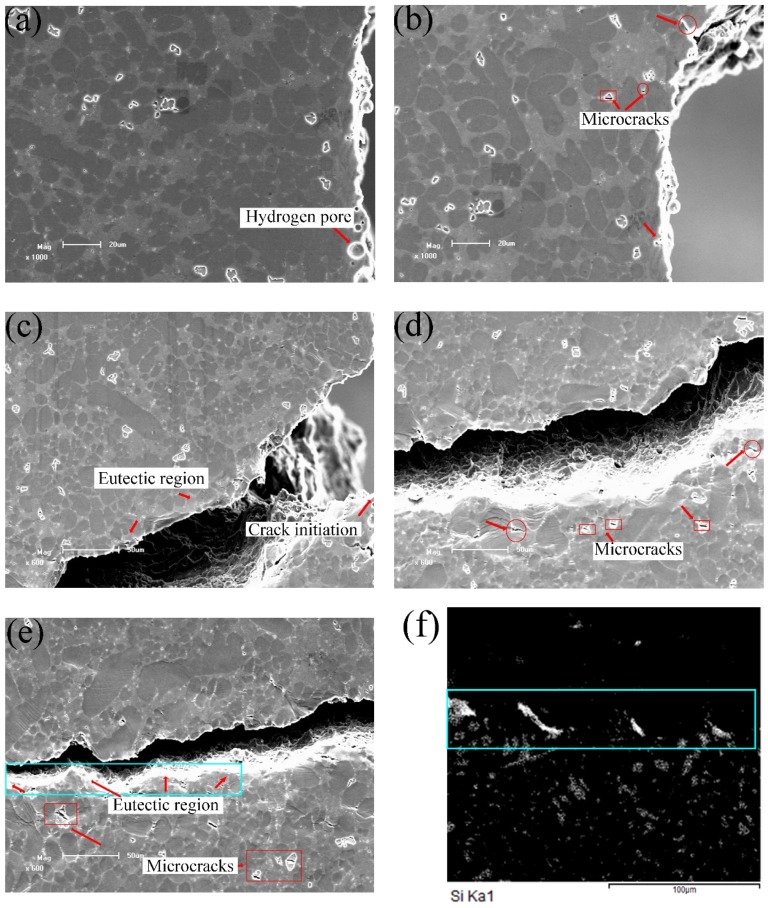
In-situ scanning electron microscopy (SEM) images of the as-cast alloys at stress/load level of: (**a**) 0 MPa/0 N, (**b**) 256 MPa/710 N, (**c**) 270 MPa/760 N, (**d**) 270 MPa/760 N, (**e**) 270 MPa/760 N, and (**f**) the EDS mapping of Si element in the rectangle region in (**e**).

**Figure 7 materials-12-02065-f007:**
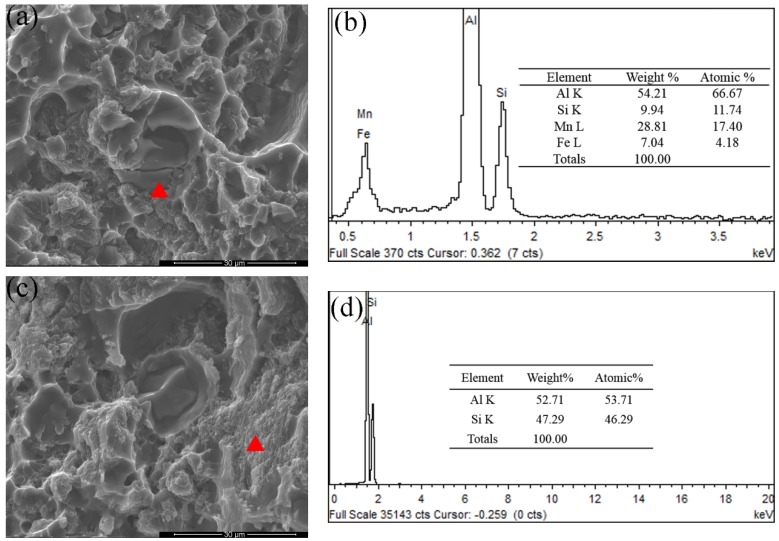
SEM fractography of the as-cast alloys. (**a**) The fracture surface containing α-Fe phases, (**b**) EDS analysis results of α-Fe phases, (**c**) the fracture surface containing eutectic Si particles, (**d**) EDS analysis results of eutectic Si regions.

**Figure 8 materials-12-02065-f008:**
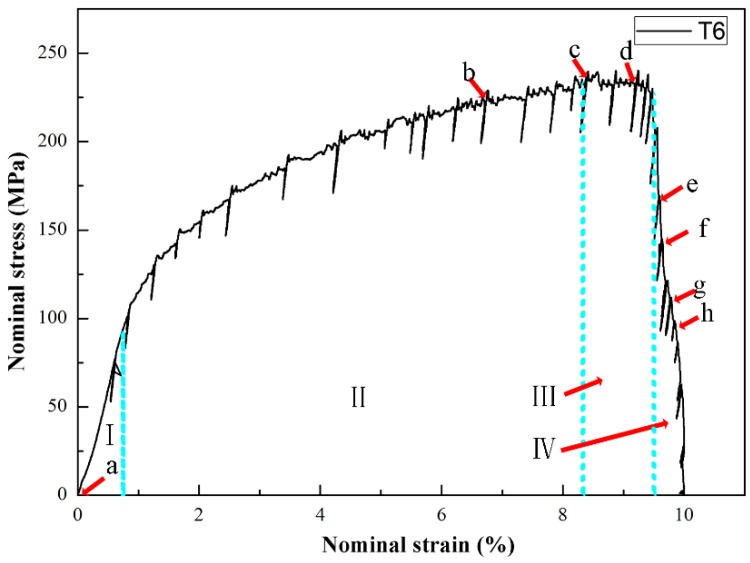
In-situ nominal tensile stress-strain curve of the T6-treated AlSiMgMn alloys.

**Figure 9 materials-12-02065-f009:**
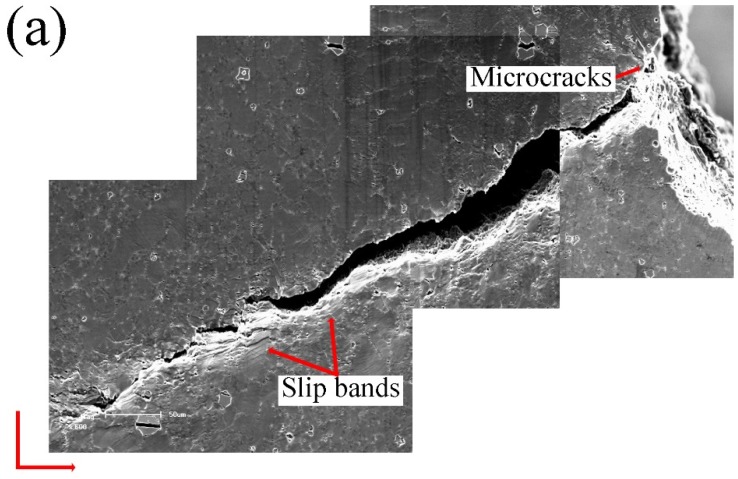
Crack propagation during in-situ tensile testing of the T6-treated AlSiMgMn alloys, showing (**a**) microcracks initiation at small subsurface pores, (**b**) voids initiation, (**c**) final crack path.

**Figure 10 materials-12-02065-f010:**
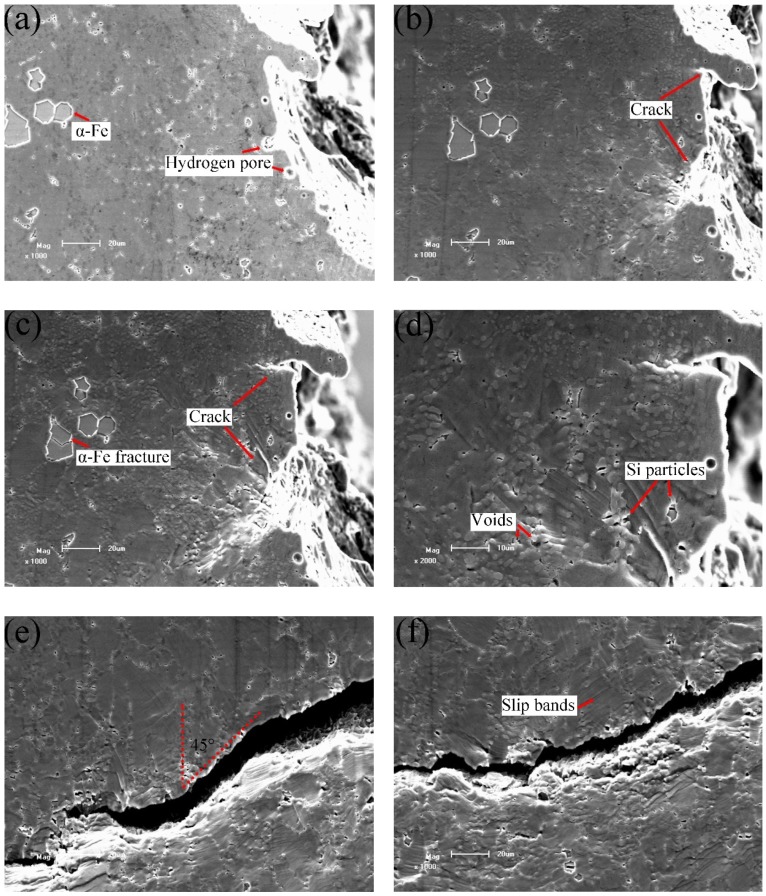
In-situ SEM images of the T6-treated alloys at stress/load level of: (**a**) 0 MPa/0 N, (**b**) 223 MPa/635 N, (**c**) 234 MPa/655 N, (**d**) 230 MPa/644 N, (**e**) 163 MPa/580 N, (**f**) 139 MPa/410 N, (**g**) 111 MPa/340 N, (**h**) 97.42 MPa/310 N.

**Figure 11 materials-12-02065-f011:**
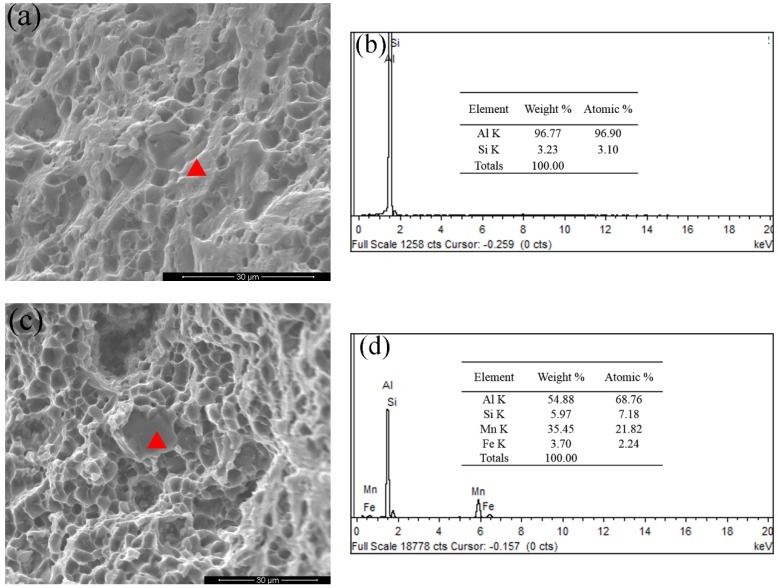
SEM fractography of the T6-treated alloys. (**a**) The fracture surface of α-Al grain, (**b**) EDS analysis results of α-Al grain, (**c**) the fracture surface of α-Fe intermetallic and (**d**) EDS analysis results of α-Fe intermetallic.

**Table 1 materials-12-02065-t001:** Chemical composition of AlSiMgMn alloy (wt. %).

Si	Mg	Mn	Fe	Cu	Zn	Ti	Sr	Al
11.37	0.20	0.58	0.10	0.009	0.012	0.075	0.017	Bal

**Table 2 materials-12-02065-t002:** Hardness and Young’s modulus in different phases.

Phase	Hardness (As-Cast) (HV)	Hardness (T6) (HV)	Young’s Modulus (As-cast) (GPa)	Young’s Modulus (T6) (GPa)
α-Al	86.8	77.2	81.2	85.1
Eutectic	145.2	98.1	87.4	79.9
α-Fe	429.6	167.2	87.2	102.1

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
