# Peer review of "Microstructure and Mechanical Properties of High Vacuum Die-Cast AlSiMgMn Alloys at as-Cast and T6-Treated Conditions"

_materials, 2019, doi:10.3390/ma12132065_

Round 1

Reviewer 1 Report

This report presents the morphology, structure and characteristic mechanical properties of high vacuum die-cast AlSiMgMn alloys in two forms, the as-cast and T6-treated conditions. The obtained results are coarse but important in the development of materials science, so the paper can be considered for publication in <materials>.

There are lots of corrections needed concerning editorial part of the paper. It seems the authors feel safer by using Past rather than the Present Tense, so generally English should be improved. The highlighted points in the text (in yellow) should be improved and/or corrected (see the pdf Enclosure).

Author Response

Dear reviewers:

Thank you so much for your comments on our paper. We have revised the manuscript carefully according to your comments and suggestions. Please find point-to-point responses below.

Response to Reviewer 1 Comments

Point 1: There are lots of corrections needed concerning editorial part of the paper. It seems the authors feel safer by using Past rather than the Present Tense, so generally English should be improved. The highlighted points in the text (in yellow) should be improved and/or corrected (see the pdf Enclosure).

Response 1: Thanks for your advice. The writing of the manuscript has been revised. The text highlighted and in yellow) in the original manuscript is modified. The modifications in blue color in the revised manuscript.

Reviewer 2 Report

The presented paper is aimed at the study of microstructure and mechanical properties of the high vacuum die-cast AlSiMgMn alloys at the as-cast 54 and T6 heat treatment conditions. The authors studied the hardness and Young modulus of the different phases with nano-indentation technique, and the crack propagation of the AlSiMgMn alloys with in-situ SEM observation. The research topic is relevant for this field of materials science. It is worth of publication, but some revisions are suggested. More specifically: 

The manuscript is sloppy and requires careful editing. 

For example, line 15, 117, 126, 130, etc.: instead of "young" should be Young; line 18,19,21: the alpha character is missing, etc. 

On lines 53-57, the aim of the work should be clearly formulated with an emphasis on the significance of the planned results. Instead, in this place is now given a brief description of the work performed. 

In the "Materials and methods" section there is no subsection describing the technique for investigating the microstructure of alloys and the equipment used for this purpose. 

line 66: Does the AlSiMgMn alloy have any brand? It is necessary to provide a reference to the standard. 

lines 110-116: These calculation formulas are better placed in the "Materials and methods" section. 

line 248: Contributions from authors should be detailed for each author.

Before the "Conclusions" section, it is recommended to add a brief description of the practical applicability of the obtained results in a foundry industrial enterprises. 

The paper can be recommended for publication after the elimination of the noted insufficiences. 

Author Response

Dear reviewers:

Thank you so much for your comments on our paper. We have revised the manuscript carefully according to your comments and suggestions. Please find point-to-point responses below.

Response to Reviewer 2 Comments

Point 1: Line 15, 117, 126, 130, etc.: instead of "young" should be Young; line 18,19,21: the alpha character is missing, etc.

Response 1: The words of "young" (line 15, 117, 126, 130, etc.) has been modified as "Young". The alpha character was missed due to the system failures and abstract were re-written. Please see the revised manuscript. (Page 1, line 11-21)

Point 2: On lines 53-57, the aim of the work should be clearly formulated with an emphasis on the significance of the planned results. Instead, in this place is now given a brief description of the work performed.

Response 2: The sentence of the work aim was added into the revised manuscript according to your opinion. (Page 3, line 58-60)

Point 3: In the "Materials and methods" section there is no subsection describing the technique for investigating the microstructure of alloys and the equipment used for this purpose.

Response 3: We have added the description of equipment for investigating the microstructure of alloys in subsection 2.2. (Page 4-5, line 85-89)

Point 4: line 66: Does the AlSiMgMn alloy have any brand? It is necessary to provide a reference to the standard.

Response 4: The brand of AlSiMgMn alloy is AA365.0 of ANSI. The reference has been added into the revised manuscript. (Page 2, line 29)

Point 5: lines 110-116: These calculation formulas are better placed in the "Materials and methods" section.

Response 5: The description of calculation formulas was re-located in "Materials and methods" section, which was introduced in section 3.2 (line 110-116) in the original manuscript and section 2.4 (line 96-104) in the revised manuscript.

Point 6: line 248: Contributions from authors should be detailed for each author.

Response 6: In the revised manuscript, we described the contributions from authors in detail. "In this work, F. L. conducted most of the experiments and analysis; R. Y. conducted some preliminary experiments and analysis; H. Z. proposed and supervised this study; H. Z. and F. S. contributed to the analysis and improved the quality of this paper." (Page 22, line 292-295)

Point 7: Before the "Conclusions" section, it is recommended to add a brief description of the practical applicability of the obtained results in a foundry industrial enterprise.

Response 7: We describe the practical applicability of the paper results in foundry industries before the conclusion according to the above opinion. (Page 20, line 270-273)
